# Enabling Sustainable Adaptation and Transitions: Exploring New Roles of a Tourism Innovation Intermediary in Andalusia, Spain

Thorsten Roser, Ksenija Kuzmina * and Mikko Koria

Institute for Design Innovation, Loughborough University London, London E20 3BS, UK;
t.roser@lboro.ac.uk (T.R.); m.koria@lboro.ac.uk (M.K.)
* Correspondence: k.kuzmina@lboro.ac.uk; Tel.: +44-(0)-2038180777

**Abstract:** Tourism is a major global and local industry creating value through services that are enhanced and enabled through intermediaries that support innovation in the sector. This exploratory case study examines the roles and activities of a publicly funded tourism innovation intermediary for small medium-sized enterprises (SMEs) and professionals in Andalucia, Spain. We note the gap in knowledge on how intermediaries may best support stakeholders in achieving resilience and sustainability in transitions in tourism service ecosystems. Building on interviews, reports, and observations, this study finds that the intermediary has successfully supported its stakeholders in enhancing their adaptability in the current service ecosystem. There is less evidence of achieving deliberate transformations towards long-term sustainability and resilience. As the intermediary is uniquely positioned at the meso-level of the regional tourism service ecosystem, this study proposes exploring engagement to cover both macro and micro-level activities to enable moving towards becoming a transition intermediary and a regional sustainability catalyst. This study furthermore proposes an expanded range of roles and activities for the intermediary to enable moving towards resilience and sustainability, while contributing to the understanding of innovation intermediaries supporting sustainability in the tourism sector.

**Keywords:** sustainable tourism; resilience; service ecosystems; innovation intermediaries; adaptation; transformation





## 1. Introduction

At its best, tourism is an important global and local industry providing livelihoods and economic opportunities to many [1]. In recognition of this and to enhance the potential of specific destinations to compete in the increasingly global marketplace, intermediary organisations that aim to enhance the marketing of destinations have emerged as key players in tourism systems [2,3]. Often with specific industry sub-sector foci (e.g., on the hotel industry), these destination marketing organisations (DMOs) have also gravitated towards the management of destinations, an evolution somewhat fraught with ambiguity due to the elusive nature of destinations and the perceived lack of control needed in active management [4]. At the same time, the digitalisation of society and services has also had a major impact on the tourism industry and on the balance of incumbent and challenger organisations [5], requiring to rethink the roles and operational strategies of intermediaries. As [4] note, the previous focus on resource management by tourism intermediaries needs to shift to also involve and engage with local stakeholders, and the environment, for the resilience of the tourism service system in the future. The sustainability of tourism services as we know them today is also highly problematic, and the industry has demonstrated high vulnerabilities to disruption. A central twin challenge for intermediary organisations in the sector is supporting stakeholders in enhancing resilience while transitioning towards long-term sustainability [6]. This implies that intermediaries need to be innovative in new

roles and activities to increasingly support their stakeholders' capabilities to deal with highly complex operational environments.

To address these challenges, public sector entities and public–private partnerships have set up dedicated support mechanisms to provide systemic developmental services, connectivity, and access to resources to enable firms to successfully navigate complex environments [7,8]. These *innovation intermediaries* are distinct from the mainly marketing-oriented DMOs as they support other organisations in building capabilities needed in delivering their missions in the ecosystem. They are often structured as dedicated, knowledge-intensive, industry-specific service organisations that diffuse and transfer technology, support management of innovation, enable access to innovation systems and knowledge networks, while intermediating through services offered [9]. While there is a recognition that these innovation intermediaries are important and central actors in ecosystems [10,11], it is less clear how well are they able to support addressing the twin challenges of fostering resilience and enabling sustainability with their stakeholders, moving towards becoming *transition intermediaries* [12,13].

In this paper, we examine a local Tourism Innovation Agency, referred to as the intermediary from now on, in Andalucia, Spain (see Figure 1). The region offers tourists a rich mix of coastal resorts, mountainous and rural destinations, and world-famous cultural sites. The local culture is distinct from other Spanish destinations due its history and proximity to Northern Africa, and the mild climate makes it a popular destination throughout the year. A recent expansion of international airports and affordable airfares has further enabled considerable growth in tourism and related economic activities. The case intermediary was set up in 2007 as a publicly funded organisation under the Ministry of Tourism, Regeneration, Justice, and Local Administration of the Junta de Andalucia (Andalucia Lab, 2023—andalucialab.org/ (accessed on 8 May 2012). accessed on 26 June 2023). Its initial offer was to support small and medium-sized enterprises (SMEs) operating in the tourism sector (some of them being DMOs, such as Just Explore (https://just-explore. com/ accessed on 20 May 2023), entrepreneurs, service professionals, and local government in Andalucia in becoming more competitive as a tourism destination, and to attract 'tech-talent', at a time when the internet was starting to affect travel industry. From the start, the scope of the supporting activities in building the capabilities of other organisations through bespoke services placed the intermediary somewhat outside the remit of marketing-focused DMOs [3], although natural bridges can be seen to exist through joint attention to services within the service-dominant logic [14]. In recent years, the offering of the intermediary has expanded to include sustainability and resilience, as noted in the recent lab-hosted workshop, entitled "Support to Spain's Tourism Ecosystem: towards a more sustainable, resilient and digital tourism" (Andalucia Lab, 2023—andalucialab.org (accessed on 8 May 2012). accessed on 26 June 2023). In this, the intermediary links up with the more recent DMO research on sustainability and sustainable tourism [15].

Much of the tourist industry offering is delivered through sets of bespoke services. However, as [4] note, the previous focus on services and associated resources has not necessarily legitimised the engagement of local stakeholders in the DMO context. However, visitors and local stakeholders can be understood as central users and co-creators of value in services in line with the service-dominant logic also in the DMO context [14]. In this paper, we maintain that for both innovation intermediaries and DMOs, the principal medium of value co-creation is services [16–18] between the producers and consumers on shared platforms [19] within tourism ecosystems [20,21]. As noted earlier, the tourism industry is currently facing major sustainability challenges, while the recent pandemic exposed significant vulnerabilities and unsustainable practices in the current industry structure and operational logic.

Through charting current and possible new roles in sustainable adaptation and transitions, our aim in this paper is to explore the potential that the tourism organisation has in becoming a transition intermediary [12], and thus we ask: How does the case intermediary currently contribute towards fostering resilience and enabling sustainability within its

tourism service ecosystem? And secondly, what potentialities exist for it to move towards becoming a transition intermediary?

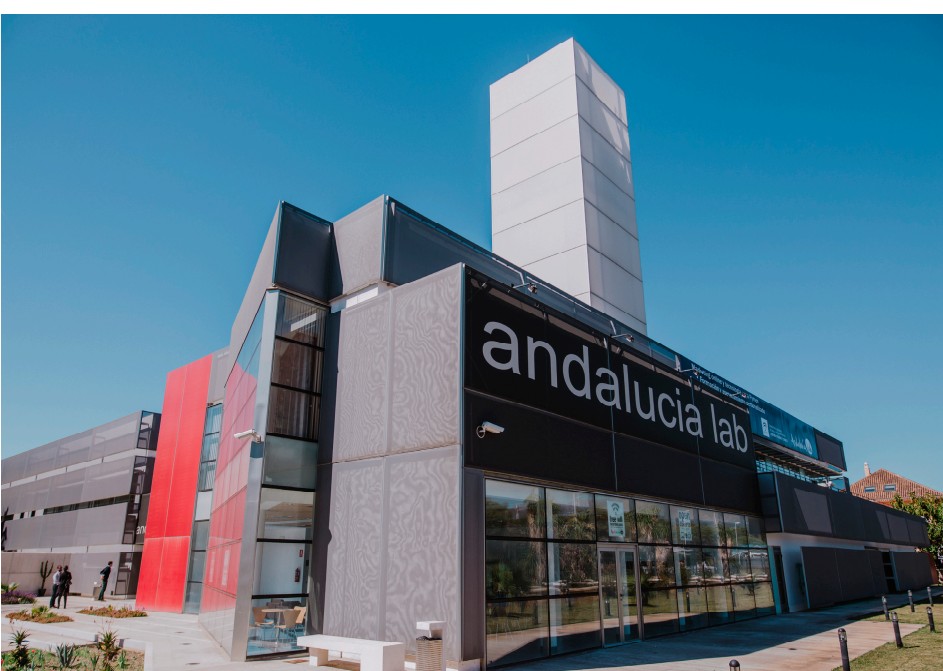

**Figure 1.** Andalucia Lab, Marbella, Spain (image by Tomasso Comazi, https://necstour.eu/good-practices/andalucia-lab-response-COVID19-crisis accessed on 10 May 2023).

In our paper, we first introduce key theoretical perspectives in tourism-related resilience, adaptability, transformation and sustainability within service ecosystems and the roles and activities of innovation intermediaries. The methodology builds on secondary sources of evidence and past participatory engagement of the authors with the intermediary and a series of interviews with key expert informants. We then examine the current roles and activities of the intermediary and use interviews to uncover the current perceptions of actors related to the tourism service ecosystem, finding that the intermediary appears to be enhancing resilience through fostering stakeholder adaptability in the current ecosystem. What is not so clear is whether the intermediary is able to support enhancing deliberate transformation, understood as the proactive and the intentional creation or shift to new or emergent developmental pathways [22] that would enable sustainability in tourism service ecosystems through service innovation on business-to-business (B2B) platforms. Based on our findings, we propose expanding the current meso-level activities to include both macro and micro perspectives to enable sustainable transitions through applying *user-centred service design approaches*. This study supports the understanding of how intermediaries could enhance resilience through further fostering adaptability and especially by enabling transformation and long-term sustainability through their offering and activities. We contribute by identifying the strategies that could help the innovation intermediaries to move towards becoming *transition intermediaries* through an enhanced adaptability and capacity for transformation to foster resilience so as to withstand market disruptions, while also providing developmental paths and measures to facilitate long-term sustainability of the industry.

## 2. Literature Review

### 2.1. Resilience, Sustainability, and Tourism

The capacity of withstanding shocks (such as pandemics, climate change, or market disruption) is based on a system's resilience [23,24], understood here as the ability to absorb disturbance and re-organise while undergoing change, whilst retaining the same functions, structure, identity and feedback [25]. Systems have the capacity to adapt and influence

resilience, but also to transform into new, more sustainable ones when socio-economic or ecological structures so demand [25]. Adaptation is the process of reducing the system's vulnerability through sensing disruptions, changing existing configurations and practices, integrating new resources, including knowledge and experience, and strengthening networks of relationships [26]. Transformation refers to shifting the development of the service systems towards new, emergent pathways or creating novel pathways [22]. Transitioning towards more sustainable socio-economic systems implies transformations in and between actors, infrastructure, technologies and application contexts [12]. To date, research in transitions has identified several approaches related to multiple levels, technological systems, strategic management and the management of transition [13]. Renewal may be motivated by major disruptions disasters or any other crisis, both forcing and enabling change [26], or it may be led intentionally through agenda-setting by participating actors, leading to change across multiple levels of society [27]. Deliberate transformation involves breaking down current resilience and building up new resilience [23], potentially involving both incremental and radical innovation [28], and ambidextrous exploiting and exploring [29], building on service innovation [30,31].

Adaptation and transformation also play a role in enhancing resilience and enabling sustainability in tourism [32,33], seen here to consist of nested and networked sets of services co-created by multiple actors in service ecosystems [34]. These socio-ecological systems operate through multi-contextual resource integration and co-evolve within their economic, social and ecological environments [35]. They are complex, adaptive, co-evolving and non-linear [24,36] and value is co-created through services around tourist experience, underpinned by changing desires, needs, and tastes [37], involving both man-made and natural capital [38]. Examining resilience and sustainability in tourism involves community development, education, health and well-being [39–41] and long-term sustainability through renewal and regeneration [42] or 'evolutionary resilience' which is increasingly supplanting 'engineering resilience' or a quick bounce-back [43]. Zolfani et al. [44] argue that narrow industry and research views on economic issues [39] and ecological/environmental impacts [45] have proven insufficient, including the view of industry as supply led, with finite resources and fixed entities, encouraging resource conservation [38].

*2.2. Innovation Intermediaries*

Building on past research in the areas of innovation and management [46,47], Howells [48] sees innovation intermediaries diffusing and transferring technology, supporting managing innovation, enabling accessing innovation systems and knowledge networks, and intermediating services through scanning information, knowledge processing, brokering, validation and commercialisation [49]. Roles as conveners, mediators and learning catalysts have also been proposed [50] as well as examining intermediation and users, while the proposed taxonomies [11,12] recognise transition intermediaries as having key roles in sustainability drives. In turn, the DMO line of scholarship views intermediaries in the tourism sector from the perspective of destination marketing [2,3], later evolving into destination management [51]. Notwithstanding the overarching marketing focus on planning, strategies, implementation and control [3], some attention has also been given to customer and user experience [52] and service-dominant logic [14], which form natural bridges between the two research streams.

In this study, we take a wide managerial view on the intermediary, building roles that enable supporting the marketing-focused DMOs in their missions. Our case intermediary supports collaborative innovation through helping to sense disruptions, revise and support evolving current configurations and operational practices, reconfigure and integrate new resources to their activities, include knowledge and experience, and strengthen networks of relationships [26,53]. To note, however, sustainability in the ecosystem level has received less attention [16,17] and researchers [26,53] argue that the resilience of tourism service ecosystems is based on social actors developing capacity to adapt and transform for systemic persistence and resource integration during periods of abrupt and gradual

change. Research [26] also identifies these actors as intermediaries exerting significant influence through their leadership and direction, facilitating supportive and interactive activities resulting in enhanced coordination and collective capacity for action at multiple levels. It has also been proposed [53] that these intermediaries can positively influence the move towards sustainable socio-technical systems through their roles and agency. These transition intermediaries [14] create new collaborative engagements linking both current and new actors, enhance skills and resource bases, while also disrupting the existing status quo [12].

### 2.3. Services and Ecosystems

Intermediaries as well as DMOs operationalise their roles and execute their activities in service ecosystems mainly through B2B services on intermediary platforms [19]. Multiple intermediaries design and deliver local/regional tourism services as the principal medium of value creation via value in exchange, value in use and value in context [15–18]. In the case intermediary context, value is seen as co-created for users, organisations, the ecosystem and society as a whole [54] between the producers and consumers and partnering DMOs. As other contexts indicate, potential benefits may accrue also in the tourism sector from acknowledging the human factor and related sensemaking as key factors in service provision [20,21]. Service innovation research to date has evidenced how engagement of users improves customer experience-driven destination management [37]. At the core is understanding but also shaping tourist experience, desires and tastes, within a value co-creation logic [17]. Host communities are also important stakeholders [38,44] within the value propositions of tourist destinations [38,55]. However, current research on stakeholder participation is mostly limited to consultation [44] and further active engagement, intermediation and more research on stakeholder participation and governance is needed. Finally, disruptions (of any kind) pose a 'wicked' problem [56] for the tourism service ecosystems as there is no single right answer in balancing current health and economic issues with those of future resilience and sustainability.

### 3. Materials and Methods

This exploratory study adopts a qualitative single case study approach [57,58] (see Figure 2) to describe and explain current intermediary roles and activities while theorising [59,60] on potential transition futures within a situated service ecosystem. This study presents a unique opportunity to understand the roles and activities of an intermediary that successfully operates in its context. The aim of the inquiry was not generalisation and prediction, but to initially describe and illustrate the issues in a specific set of circumstances and then develop new frames of reference. We broadly followed the process of an abductive thematic network analysis where research explores the associations between emerging themes inherent in the data at hand [61,62]. Building on sensemaking and user-centred approaches from service innovation research [63–65], we identified and analysed future service opportunities for the intermediary. We recognise that multiple possible 'truths' can co-exist within the same data [66] and are iterated between the concepts presented in the literature and the empirical evidence in the data; in this way we are aiming to make sense of the current realities and perspectives, while exploring the future pathways.

| Primary and Secondary Data Collection | | | | Abductive Thematic Network Analysis |
|---|---|---|---|---|
| 1. A posteriori based on the experience with Andalucía Lab over the 10 years. | 2. In-depth (1 h) Interviews × 10 with regional experts. | | 4. 15 Andalucía Lab reports | 1. Understanding current realities and perspectives. |
| | 3. Design Sprints × 2 | | 5. Two video recording of corporate group presentations | 2. Generative themes exploring futures. |

**Figure 2.** Data collection and analysis framework.

This study combines multiple primary and secondary sources of data to add depth and breadth in describing and analysing the scope of the activities of the intermediary [67,68] (see Figure 2). The value of mixed methods for a single case study has been widely recognized, as it allows for compelling convergence of evidence [57]. Exploratory indepth interviews were conducted jointly by two researchers with ten regional tourism SME experts in 2020. Due to the pandemic, these interviews were carried out online only. However, familiarity with the intermediary enabled purposeful sampling (see Table 1), facilitated by the intermediary through participants recruitment and in-kind support. The selected interviewees (managing director or equivalent level tourism professionals currently operating in the service ecosystem, typically in DMOs) had previous experience with the intermediary's services and expressed a personal interest in participating in our study.

**Table 1.** Interview participants.

| Participants | Responsibility | Organisation | Years Active |
|---|---|---|---|
| Participant 1 | Director | Online Destination Management Company; covers all of Spain | Since 2014 |
| Participant 2 | Managing Director | Andalucia Lab | Since 2007 |
| Participant 3 | Managing Director | Team Building and Event Management Firm; covers Andalucia, Barcelona and Seville | Since 2000 |
| Participant 4 | Operations Director | Bespoke Travel and Private Tours, Andalucia | n/a |
| Participant 5 | Managing Director | Bespoke Travel and Private Tours, Andalucia | n/a |
| Participant 6 | Managing Director | Tourist Apartment Management Company; Andalucia, Seville and Malage | Since 2009 |
| Participant 7 | Consultant, Big Data Analytics | Analytics and Innovation for the Tourism Industry | Since 2010 |
| Participant 8 | Managing Director | Event Services Firm | Since 2011 |
| Participant 9 | Statistics and Market Researcher | Tourism Statistics and Market Research for Andalucian Regional Government | n/a |
| Participant 10 | Owner | Digital Marketing Platform for Travel Agencies, Operators, Cruise Companies, and Hotel chains | Since 2000 |

In addition to uncovering activities, the interviews identified developmental needs and challenges in moving towards resilience and more sustainable tourism futures. Interview data were transcribed verbatim and anonymised using a standardised transcription code with the express permission of the research participants. The transcripts were proofread for accuracy by two independent researchers. Using ATLAS.ti 9.0, the structured method involved first and second cycle codings by two independent coders [57,62]. Our qualitative analysis was based on intuitive reasoning [61] allowing us to develop hypothetical explanations and to associate the data with frames of reference for further investigation [69,70]. During the first cycle of analysis, the data were screened, and a codebook was developed. The creation of codes was informed by the data and academic literature. Codes were then assigned to paragraphs and distinct sections of the data at hand. During the second cycle coding process, initial codes and analytic memos were revised and re-applied with a view to develop basic, as well as higher order analytical themes that would highlight critical issues, meanings and linkages between different data. Both coding cycles involved two independent researchers who would code and analyse the data independently and then discuss and consolidate their findings to provide a coherent analysis and interpretation of the data.

The second main method of primary enquiry was related to participatory observation [71]. Since 2010, one of the researchers has closely followed the incremental development of the intermediary's services, being an early stage user of the facilities. In addition, while the early focus on the digitalisation of the industry has profiled the intermediary in the forefront of service innovation, the intermediary staff engaged in two design sprints [72]

with the researchers, developing new services in the context of university–industry collaboration [73] (See https://www.lborolondon.ac.uk/cpshow21/andalucia-lab-1/ accessed 10 May 2023). The main secondary data sources included a series of fifteen recent key Andalucian travel industry reports (2000–2018) with socio-economic and sustainability related contextual data as well as European and regional travel statistics (via statista.com (accessed on 8 May 2012). accessed 10 May 2023), two video recordings of group discussions and corporate presentations by industry experts.

## 4. Results

### 4.1. The Intermediary Roles and Activities

Our research shows that the services offered include data and information sciences support, research, training and consulting, and facilities for co-working, exhibitions and events. The B2B offering involves tourists and local citizens only as indirect beneficiaries. The intermediary is seen as a pioneering local 'Tourism Innovation Centre'. Equipped with modern office infrastructure, exhibition space and event facilities, it attracts members and visitors, provides co-working spaces, meeting rooms, training facilities, lecture theatres, and assembly and exhibition spaces. From the start, the intermediary's key objectives included knowledge sharing and networking between technology providers, entrepreneurs, tourism professionals and DMOs. Since 2012, a further aim has been to attract and incorporate specialist 'Travel Technology Companies' to strengthen the regional tourism ecosystem. The agency provides specialist consulting services and training in digital capabilities, skills, and entrepreneurship to foster long-term competitiveness. Collaborating with local municipalities, it supports and promotes regional destinations, businesses, entrepreneurs, and independent service professionals. Table 2 summarises the current roles and service offering of the agency based on the data from observations, reports, and recordings, that building on the intermediary taxonomies proposed to date [11,48].

**Table 2.** Current roles and service offering of the Andalucian Tourism Innovation Agency; data analysis from observations, reports, and recordings.

| Generic Aims of Services | Current Intermediary Roles and Service Offering |
|---|---|
| **Diffusing knowledge and enabling technology transfer**<br>Ensuring tech feasibility through scanning, evaluation, foresight, and road-mapping of technology | **Expert in tech transfer**<br>Hosting a 'demo lab' exhibition space to present new technologies to users<br>Digital showcasing provides content related to innovations and the latest trends in digital technology for tourism<br>Attracting visitors and exhibitors to the facilities to showcase technology demonstrations, simulations or to provide workshops that introduce new technologies to future tourism professionals |
| **Systems and networks**<br>Engaging networks in storing, and sharing knowledge, and bridging to external resources | **Bridge to resources**<br>Creating opportunities for ideation and networking<br>Disseminating activities to learn about new technologies and their application in tourism (working with schools, research institutes and universities)<br>Launching an international hub for tourism<br>Hosting an open café area as a meeting space for the entrepreneurial community<br>Delivering knowledge transfer session to a community by supporting entrepreneurs, independent professionals, and destinations in tourism |

**Table 2.** *Cont.*

| Generic Aims of Services | Current Intermediary Roles and Service Offering |
|---|---|
| **Intermediation Services**<br>Educating stakeholders through identifying, structuring, and delivering learning activities | **Catalyst in learning**<br>Organising workshops, business know-how and training tailored to the needs of industry professionals<br>Hosting master classes by professional experts<br>Organising meetings to foster cooperation and business between tourism professionals, tourism entrepreneurs and technology providers<br>Providing advice to professionals via a 10-week programme of specialist consulting sessions |
| **Innovation management**<br>Enhancing business viability through exploring, identifying selecting, negotiating, marketing, and exploiting market opportunity | **Broker of development**<br>Diffusing knowledge and information related to online marketing<br>Organising conferences related to specific topics, current affairs and new trends to encourage innovation<br>Hosting a co-working space for circa 70 companies<br>Providing space for events, workshops, networking, a large 200-person lecture theatre, and access to the services of local tourism professionals and entrepreneurs |

Source: Authors, with adaptations from [11,48].

### 4.2. The Tourism Service Ecosystem

Our interviews uncovered further insights from the stakeholders of the service ecosystem. Stakeholders do also recognise the need for transformative support in transferring expertise beyond a 'digital update', for example, in implementing new health policies and safety protocols or seeking an intermediary to serve as a translator of the measures established by the government for the travel industry. While recognising the opportunity, the intermediary has been reluctant to engage with the sustainability agenda independently: "*The private companies are not going to have a sustainable approach, at least at the beginning, if you are not dealing with a frame that the government are setting up*" (Participant 2). While the current portfolio of activities of the intermediary appears to support the adaptation processes of stakeholders through access to the current technology and infrastructure, it remains unclear how new up-and-coming areas of knowledge could be incorporated in the offering, and it is also unclear which areas should be taken into consideration in the future.

In terms of supporting pathways to bridging resources, the government is challenged in direct collaboration, as working with SMEs (this evidently includes DMOs) is a "*difficult task now, first of all, because we are not so used to working with SMEs*" (Participant 9). Opportunity is seen in the intermediary acting as a driver for motivating and enabling SMEs to implement and evaluate government's existing and future sustainable management policies, such as Green Deal Plans set out by the European Union, in close collaboration with local governments and through the utilization of digital technologies. There is recognition that the intermediary is well-positioned to undertake this coordination role as they have an understanding and a history of working with SMEs, and as an agency representative emphasises: "*we have been working on dealing with the private sector for a long time. So, we have this experience . . . it's easier for us to understand what is happening in the market*" (Participant 2). Interviewees indicated that adaptation and resilience appear to have been strengthened to date by the activities of the intermediary. What remains somewhat open is understanding that future knowledge and resource need to bridge resources and create novel connections between actors.

Voiced achievements also exist for the intermediary in catalysing learning due to the activities within the ecosystem. The learning activities, events and meetings have enabled the ecosystems actors, including DMOs, to (re-)position themselves to withstand the impact of external shocks. However, a perceived engagement gap has also been identified by the service providers and the intermediary in extending offering to the micro level of

users and consumers. Weak feedback loops do not enable service or policy development that is underpinned by and shapes tourist desires and tastes whilst utilising on-demand resource management to achieve more sustainable tourism. Response to market disruption was to entice tourists with lower prices, rather than to address issues of, e.g., trust and confidence, which they see as an important part of the 'new experience': " . . . *Travel agents get together and tell us what you can offer, which prize which experience [. . . ] which discount, which voucher? [. . . ] I think it is not a question of putting out there a bigger demand and overwhelming the traveller. It's more about finding a way to [. . . ] get and feel confident.*" (Participant 1). Yet, stakeholders emphasised the transformative need to see tourists not as rational consumers with a pre-defined set of needs, but as users who experience services in-use, including pre-service decision making: "*We need to face that flying is going to be affected. And not just because of the costs, but also because people maybe don't feel comfortable being in a full flight for the holidays . . . this is something that is now in the minds of all of us, as citizens and as potential tourists*" (Participant 9). While perceived as an emerging trend, the 'responsible client' is also seen as a desirable customer for the tourist industry, however, little thought has been vested on how the sector can gain user and consumer-centred knowledge and educate for or support this perspective.

Overall, the industry struggles to position itself as an innovator targeting transformation for sustainable and resilient future and brokering development in this area emerged as an identified need. The intermediary has been able to foster development in positive ways in the past, but the latest market disruption induced by COVID-19 exposed deep vulnerabilities and made stakeholders and participating DMOs aware of the need to digitalise their services even further with many stakeholders indicating they " . . . *used this quiet time to innovate and to update, and to prepare things . . . to refresh the website, to create itineraries . . . *" (Participant 1). The intermediary has been able to support its stakeholders in adapting to new service requirements: "*We are trying to help small companies . . . now that they are aware about, that it's very important to digitise their businesses . . . now and it is more evident than ever*" (Participant 2). Overall, this is indicative that a core offering of the innovation intermediary continues to be well-aligned with the adaptive demands and needs of their stakeholders. Yet, the findings suggest that there is a narrow view by the industry on what sustainable tourism is or can be. The narrow view also extends to developing business models in the area, and developing innovative models for sustainable tourism in Andalucia has been discussed as another transformative opportunity area for the intermediary. Business models based on small-scale tourism are seen by the service providers to provide more resilient experiences, needing adaptation and incremental innovation: "*We offer a very customised service for independent travellers and small groups . . . I think the business model we have in planning is the trend that is going to be next. So, I don't think we need to change anything*" (Participant 3). While this narrow position is further supported by the view that small group tourism may well become a safer and more desirable option for the consumer in the future, it fails somewhat to address the nature of future activities and no clear strategy expressed by the intermediary or other stakeholders exists on how to engage in a transformative fashion with tourists to re-appraise and re-design the service offering. A view on sustainability was expressed as a set of behaviours embodied in individual tourists becoming "responsible clients" prior to arriving at the destination: "*People that know when they are coming to the beaches, they are not going to leave plastic on the sand . . . So, I think . . . we have to think about responsible people . . . think there is a trend of more responsible client*" (Participant 2). Engaging in transformational development implies rethinking how the intermediary acts not only with current business stakeholders, but also with the users and consumers of tourism services.

## 5. Discussion

In this paper, we have examined how the case intermediary contributes towards fostering resilience and enabling sustainability within its tourism service ecosystem, while aiming to explore the potentialities that exist for it to move towards becoming a transition

intermediary. Analysing the roles and activities confirms the organisation's profile and role as an intermediary with a top-down mandate and normative positioning [74] in agenda setting, exploiting, developing capabilities, and supporting consensual continuity. As a meso-level operator, it creates value for its B2B stakeholders, such as DMOs, in supporting their adaptation to volatile and changing operational environments, primarily through collaboration within digital technology and tourism. The current roles and activities enable adaptive access to resource, knowledge, and expertise diffusion via training, co-working and events through transferring expertise, bridging resources, catalysing learning and brokering development. This has enhanced adaptability and thus resilience, within the current techno-economic rationale and roles.

However, there was a lack of evidence of impact of the intermediary successfully supporting deliberative transformation and of the positive systemic transitional impact on the tourism service ecosystem. Within the current tourism service ecosystem, there appears to be a need to further the engagement with multiple other intermediaries and DMOs to rewire the local and regional tourism ecosystem, as well as to redefine the role the intermediary could play to enable and expand this network. Furthermore, the current meso-level positioning remains too detached from both macro and micro-level decision-making, resulting in an inability to influence and interact both policy issues, and user and consumer desires and behaviour. Weak macro-level feedback loops do not encourage multi-level policy development, enhance tourism governance systems or experience-shaping on a wider front. The lack of micro-level focus also creates engagement and collaboration gaps between the micro-level actors, tourists, users and customers of services, local communities and citizens who have no current role or voice in developing transformative innovation and new offering as lead users [75]. There is a notable lack of attention on brokering the development of new services through user-driven innovation [30] and in creating new knowledge feeds into the system. As an example, while the intermediary involves schools and universities in demonstrating their work to inspire participation in the tourism–tech industry, there appears to be little engagement or dedicated outreach programmes. The potential transformative role of the intermediary may also be held back by the current emphasis on technology that pays little attention to catalysing learning in the area of human experience and desirability.

*Strategies for Enabling Transformative Resilience and Sustainable Futures*

While the current focus on adaptation enables the intermediary to leverage existing capabilities and exploit them to the benefit of the ecosystems, the key challenge, and a wicked problem [56] for the intermediary and its stakeholders and participating DMOs concerns enabling deliberate transformation to enable sustainability [22] within its service ecosystem. This does not imply that the current offering could and should not be developed further. On the contrary, developing existing service offering further enhances *adaptation*, and the intermediary could aim to become better and more inclusive within its current scope of activities on the meso-level; in other words, exploiting still furthers the existing capabilities, connections and modes of operation, but now involves a transition perspective. This implies that the intermediary continues to transfer expertise to its stakeholders in the service ecosystem through, say, scanning, evaluating, foresight, and road-mapping technology. Similarly, bridging resources can continue to involve capturing, storing, and sharing knowledge and connecting ecosystems stakeholder to the resources. Adopting a transition perspective is needed, implying new purposeful positioning within an existing structure.

That said, engaging in *transformation* recognises and brings about new requirements and uncertainty. This implies connecting proactively with the wider socio-ecological systems which contain the tourism service ecosystem: shifting attention, exploring new modes of operation and bridging to both the macro and micro-level actors in the ecosystem. To proactively shift to a more exploratory mode of operation, it would require building up ambidextrous capability in the organisation [29,76,77] that would allow the intermediary

to exploit the existing ecosystems, while exploring new transition dimensions. This would mean extending the roles and activities of the intermediary to new areas that would allow it to exploratively develop new solutions and future pathways, while keeping a steady aim on sustainability targets. We argue that this requires expanding the role and activities of *catalysing learning* to include multi-level and transition perspectives and knowledge in identifying, structuring and delivering learning activities. We also note that the role and activities of *brokering development* requires expanding the perspective in exploring, identifying, selecting, negotiating and exploiting market opportunities to consider and embed transition strategies and viewpoints. Potentially, the intermediary can become an innovation catalyst, leader and accelerator of user-driven tourism innovation, supporting co-creation practices between local businesses, government, and tourists and citizens. However, we argue that becoming a catalyst requires adopting of two additional intermediary roles, as a macro-level *transition influencer* and a micro-level *translator of meaning*.

On the macro-level, there are signs of travel firms transitioning from mass-tourism-oriented (i.e., large group) towards more distinct and bespoke small-group tourism [45]. Current service and business models will thus require radical rethinking, and emphasis on 'luxury', 'wellness' and 'agritourism' may also create demands for 'safety, 'authenticity' and 'responsibility'. Trends might move from 'fun and party' towards 'relaxation and wellbeing', including 'authentic' cultural experiences, especially in an ageing society. Intermediaries would also need to address the challenges of time-limited lifecycles and popularity of tourism destinations [78] in the progress towards sustainable futures [79]. Positioning the intermediary as a transition-minded convenor for joint multi-actor initiatives implies extended timeframes and complex operational environments and potentially becoming the owner and champion of a transformative change agenda within socio-ecological ecosystems. We argue that this role is one of a *transition influencer*, convening actors, interpreting and harmonizing (e.g., policies to local contexts), counselling and at times enforcing the sustainability agenda, building on current expertise and technology. This requires influencing the setting of new priorities, reconciling local, regional, national and global agendas, as tourism policies impact the success of ventures and destinations through new services at the intersection of big data and public policy. Addressing more than economic factors implies moving away from traditional resource-based perspectives [80], while balancing social, economic and environmental considerations [81]. As such, the intermediary is potentially well placed to develop and provide services that support stakeholders in local and regional transitions towards these transformative and sustainable practices [82].

On bridging to the micro-level, we note the extensive opportunity that exists in applying service design methods, approaches and tools [13,18]. User experience and customer journeys underpin understanding desirability and local user-centred views within service ecosystem development [64]. This requires the intermediary to adopt the role of a *translator of meaning* in discovering, defining, and developing user needs, desires and experience to curate new sustainable service offering. This creates opportunities to bring technologies closer to users and expands the role of tourists as users to inform continuous improvement of existing services. It also allows engaging the users and consumers as lead innovators in new solutions that can make a difference [75]. Enhancing service design and delivery capabilities would appear to be a pre-requisite in building ambidextrous capabilities [83–87], as the same approaches, tools and methods can be applied to deliberate and transformational change and development.

Moving our case intermediary towards becoming a *transition intermediary* thus involves revising and expanding the existing roles and activities of the intermediary in the service ecosystems. We argue that moving from adaptation to transformation enables a significant leap in this direction, and by adding two new identified roles while also reconfiguring existing one allows us to update key roles and activities within a multilevel perspective. In Table 3, we summarise the extended roles and activities of the intermediary.

**Table 3.** Transition intermediary: drivers, roles, and activities in resilient and sustainable tourism service ecosystems.

| Generic Aims of Services | Drivers in Transformation | Key Roles of a Transition Intermediary | Intermediary Activities |
|---|---|---|---|
| Diffusion of knowledge and technology transfer | Ensuring tech feasibility | Expert in tech transfer | Scanning, evaluation, foresight, and road mapping of technology |
| Systems and networks | Engaging networks | Bridge to resources | Capturing, storing, and sharing knowledge and bridging to external resources |
| Intermediation services | Educating stakeholders | Catalyst in learning | Identifying, structuring and delivering learning activities |
| Innovation management | Enhancing business viability | Broker of development | Exploring, identifying, selecting, negotiating, marketing, and exploiting market opportunity |
| Transition management | Enacting sustainable futures | Transition influencer | Convening, interpreting, harmonizing, counselling, and enforcing the sustainability agenda |
| User engagement | Enabling human desirability | Translator of meaning | Discovering, defining, developing and delivering on user needs, desires and experience |

Source: Authors, with adaptations from [11,16,48].

## 6. Conclusions

In this paper, we examined the roles and activities of a Tourism Innovation Agency, an innovation intermediary in Andalucia, Spain. The evidence indicates that the organisation supports enhancing the adaptability of its stakeholders but falls short of successfully supporting deliberative transformation and thus long-term sustainability and positive systemic impact on the sustainability of the tourism service ecosystem. The research highlights the importance of further supporting the adaptation processes of stakeholders while aiming for transformative approaches to enable resilience and sustainability. This is seen to require significant revision in the current roles and activities of the intermediary while also expanding new roles as proactive influencers and translators between macro, meso and micro-layers of the service ecosystem.

Based on our findings, we propose expanding the current meso-level activities to include both macro and micro perspectives to enable sustainable transitions through applying *user-centred service design approaches.* This study supports the understanding of how intermediaries could enhance resilience through further fostering adaptability and especially by enabling transformation and long-term sustainability through their offering and activities. We contribute by identifying the strategies that could help the innovation intermediaries to move towards becoming *transition intermediaries* through an enhanced adaptability and capacity for transformation to foster resilience so as to withstand market disruptions, while also providing developmental paths, strategies and evident managerial implications to facilitate long-term sustainability of the industry.

The outlined approaches create opportunities for the intermediary to re-position itself within the service ecosystem, with the potential to become a key hub and influencer in this regard. This would, however, require expanded capabilities in both exploitation and exploration. We suggest that applying service design and delivery approaches, tools and methods would support this transformation process through enabling user-driven service innovation. These topics would also form a natural direction for further research. Recognising the limitations in generalisability, inherent in qualitative single case studies

that have an exploratory nature, the study nonetheless contributes to our understanding of how tourism innovation intermediaries can support resilience enhancement through further fostering adaptability and especially enable deliberate transformation and long-term sustainability through expanding their offering and activities.

**Author Contributions:** Conceptualization, methodology, formal analysis, investigation, resources, data curation: writing—original draft preparation; writing—review and editing, T.R., K.K. and M.K.; visualization, M.K.; supervision, M.K. and K.K.; project administration, T.R.; All authors have read and agreed to the published version of the manuscript.

**Funding:** This research received no external funding.

**Institutional Review Board Statement:** The study was conducted in accordance with the Declaration of Helsinki and approved by the Institutional Review Board (or Ethics Committee) of Loughborough University.

**Informed Consent Statement:** Informed consent was obtained from all subjects involved in the study.

**Data Availability Statement:** The data presented in this study are available on request from the corresponding author. The data are not publicly available due to data protection.

**Acknowledgments:** We would like to express our deepest gratitude to the regional experts and entrepreneurs involved in this study, who supported this research with their time and expertise during a challenging time for the tourism sector. We thank the team at Andalusia Lab for their kind support and cooperation with this research. Particularly, José Luis Córdoba Leiva and Paloma García-Delgado who have been invaluable in enabling the recruitment of the research participants. We also thank Pedro Heredia Medio for his local knowledge and support.

**Conflicts of Interest:** The authors declare no conflict of interest.

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
