# Peer review of "Enabling Sustainable Adaptation and Transitions: Exploring New Roles of a Tourism Innovation Intermediary in Andalusia, Spain"

_tourismhosp, doi:10.3390/tourhosp4030024_

Round 1

Reviewer 1 Report

Dear Authors, 

just few things to be revised.

I would suggest to revise the References. In particular some of them are not complete, please see the numbers 13, 44, 63 and 81. 

Number 67 and 77 are including in total 4 papers. 

I would suggest to specify, if there are not problems, the original name of the Local Tourism Innovation Agency, at least just once to allow the reader the opportunity to see its website.

Best regards

Reviewer 2 Report

Tourism is an important industry and income for many countries. Moreover, tourism services provide a significant number of jobs. The tourism industry has suffered greatly from the COVID-19 pandemic and needs solid support to return to pre-pandemic levels. For this reason, all activities that attempt to support sustainable tourism at various levels (local and national) are essential.

This study presents such perspectives, emphasizing the importance of close relations between tourists, service providers, intermediaries, and politicians. The offered example from Andalusia shows possible ways to look for solutions to improve tourist services and reduce the negative impact of tourism on the natural environment.

I congratulate the authors on a good research idea and its reliable implementation.

Author Response

We would like to thank the reviewer for their valuable time and positive comments  about this manuscript.

Reviewer 3 Report

Dear authors, this paper provides an interesting topic, but I recommend you add some suggestions to enhance the quality of paper:

1º Authors must follow the journal's rules in the reference section.

2º. Authors wrote in the title:"Enabling resilience and sustainability in a service ecosystem" which industry or sector: hospitality, museums, restaurants, ..., Would be convenient to change the title, please. Tourism provide different activities and you need to be more specific in the entire paper. For example, authors say: "This exploratory single case study examines the roles and activities of a publicly funded Tourism Innovation Intermediary in Andalusia, Spain. We note the gap in knowledge on how intermediaries may best support stakeholders in achieving resilience and sustainability in transitions in tourism service ecosystems." I did not see in which activity of tourism industry reach a good resilience and sustainability.

3º. The abstract section should be considerably improved and this tackle only: The main objectives, methods, results, and contribution, please.

4. Authors must understand that you are facing a paper related to innovation and resilience at tourism industry and supported by technologies. For this reason, you have to implement updated studies in this research. We are researchers and we are in 2023. Readers must know the last information about this topic, authors and studies  cited in 19179, 1998, 2006, 2010, 2011.. are very old studies for this research because today the tourism industry have totally changed since then. I recommend you add relevant authors like:

Florido-Benítez, L. (2022), "The impact of tourism promotion in tourist destinations: a bibliometric study", International Journal of Tourism Cities, Vol. 8 No. 4, pp. 844-882. https://doi.org/10.1108/IJTC-09-2021-0191

Chaigneau, T., Coulthard, S., Daw, T. M., Szaboova, L., Camfield, L., Chapin III, F. S., ... & Brown, K. (2022). Reconciling well-being and resilience for sustainable development. Nature Sustainability5(4), 287-293.

Meyer, C., Gerlitz, L., & Klein, M. (2022). Creativity as a Key Constituent for Smart Specialization Strategies (S3), What Is in It for Peripheral Regions? Co-creating Sustainable and Resilient Tourism with Cultural and Creative Industries. Sustainability14(6), 3469.

Bethune, E., Buhalis, D., & Miles, L. (2022). Real time response (RTR): Conceptualizing a smart systems approach to destination resilience. Journal of Destination Marketing & Management23, 100687.

Philipp, J., Thees, H., Olbrich, N., & Pechlaner, H. (2022). Towards an ecosystem of hospitality: The dynamic future of destinations. Sustainability14(2), 821.

5º. Authors need to implement Destination Marketing Organisations (DMOs) in keywords and subsection in the literature review because they manage tourism sustainable development plans in Andalusia tourism destinations and their provinces.

6º. Authors should explain why you have worked in this paper and show the gaps in this topic which authors tackle in this paper and Andalusia territory. Moreover, you must explain B2B, I know this Business-to-business but some readers not.

7º. Authors wrote this: "In this paper, we examine a local Tourism Innovation Agency – referred to as the intermediary from now on – in Andalusia, Spain. It was set up in 2007 to offer support to Small Medium-sized Enterprises (SMEs), entrepreneurs, service professionals and local government in becoming more competitive as a tourism destination, and to attract ‘tech talent’, at a time when the internet was starting to affect travel industry (Line 47-51). This sentence need to be supported by Andalusia  local or regional government as author and reference, please.

8º The introduction section is very long, I recommend authors address the most relevant information and updated studies and the main goals which stage the truth idea of this research, please. For instance, authors write in the line 84: "Based on our findings we propose expanding the current meso-level activities to include both macro and micro perspectives to enable sustainable transitions through applying user centred service design approaches. The study supports  understanding of enhancing resilience through further fostering adaptability but especially enabling transformation and long-term sustainability through their offering and activities. We contribute through identifying strategies that exist for innovation intermediaries to move towards becoming transition intermediaries through an enhanced adaptability and capacity for transformation to enable resilience to withstand market disruptions, while providing developmental paths and strategies to enable developing long-term sustainability of the industry" This paragraph must be localised in results and conclusion sections.

9º. Authors need to reconsider the structure of literature review's subsections. For instance, 2.3  Services and ecosystems. This is very general, you must be more specific in which activity of tourism industry. Another example: 2.2. Innovation intermediaries. Where in communication and marketing strategies, promotion campaigns, new technologies... Me and future readers have to know this information.

10º. 2.1 Resilience, sustainability and tourism subsection. Where in air transport, beaches, restaurants, hotels, I did not see it.

11º. The methodology has to be explained better, for example, which methods, variables, updated studies used authors to support your research, please. For instance, authors say in the line 195: "The study combines multiple primary and secondary sources of data to add depth and breadth in describing and analysing the scope of the activities of the intermediary" Why authors used primary and secondary data for this topic research. Used authors a systematic review, a specific software or variables, etc...

12º. This paper has to display a map location where this was done. Indeed, authors should show real examples of resilience and sustainability in service ecosystem in Andalusia or another tourist destinations. This helps to readers to understand the contribution of this research.

13º. Authors must explain better this: Using ATLAS.ti 9.0, the structured method 207 involved first and second cycle coding by two independent coders [68]. And supported by updated studies which used this tool, please. So we can compare results from different point of views.

14º. What dataset used authors to support your results: INE; IECA; EUROSTAT,....

15º. Authors have to add the link and reference: The intermediary is under the regional Ministry of Tourism, Regeneration, Justice and Local Administration of Andalusia.

16º. Authors say: "Our interviews uncovered further insights from the stakeholders of the service eco system" I need to know the date, locations and sample of interview, please.

17º. Authors say: In terms of supporting pathways to bridging resources, the government is challenged 260 in direct collaboration, as working with SMEs is a “difficult task now, first of all, because we 261 are not so used to working with SMEs” Which government: Andalusia, Spain, Málaga, Seville, I do not know and see it.

18º. Authors write in the line 317: “We offer a very customised 316 service for independent travellers and small groups… Where in OTAs, DMOs official websites, hotels, public and private transport....

20º. Authors say in the line 332: In this paper, we have examined how the case intermediary contributes towards fostering resilience and enabling sustainability within its tourism service ecosystem. I did not really see it in the paper. The entire paper provides a general information, and authors did not tackle the main objectives of this research. Indeed, this "study" add nothing new to the literature review and tourism industry. For instance, authors say: There is a notable lack of attention on brokering the development of  new services through user-driven innovation (line 359) in which activity of tourism industry, please.

21º. Results and conclusions sections must be considerably improved, because these did not show to respond the objectives of research.

22º. The conclusion section is very short. Furthermore, authors have to implement theoretical and managerial implications, limitations and future research subsections after the conclusion section. I am confident that authors have a lot of information for these subsections.

Round 2

Reviewer 1 Report

Dear Authors, 

thank you very much for your reply.

Best regards

Author Response

Thank you, 

Dr. Ksenija Kuzmina (and co-authors) 

Reviewer 3 Report

Dear authors, 

I did not see significant changes. Indeed, authors did not tackle the majority of muy recommendations. Why?

This paper adds nothing new to the tourism literature and scientific community. Furthermore, I recommend authors, you need to seek more information related to this topic, and know how public organizations and private companies operate in the region of Andalusia. Everything is not theory.
